# The association between cadmium exposure and lung cancer risk: A protocol for systematic review and meta-analysis

Lin Chen, Min Zhou, Dingliang Lv, Shuiwei Qiu[ID]*

Cardiothoracic surgery, the Quzhou Affiliated Hospital of Wenzhou Medical University, Quzhou People's Hospital, Quzhou, Zhejiang, China

* Qiusw3000@163.com

## Abstract

### Background

Cadmium (Cd), a toxic heavy metal, is classified as a human carcinogen by the International Agency for Research on Cancer (IARC). Epidemiological studies suggested a potential link between cadmium exposure and lung cancer risk, but evidence has remained inconsistent, particularly regarding differences between occupational and general populations. This protocol will outline a systematic review and meta-analysis to quantify the association between cadmium exposure (via blood, urine, hair, or environmental samples) and lung cancer risk.

### Methods

This study protocol will follow the Preferred Reporting Items for Systematic Reviews and Meta-Analysis Protocols (PRISMA-P) guidelines. A comprehensive search of PubMed, Embase, Web of Science, Medline, and Cochrane databases will identify observational studies published from inception to December 2024. Two reviewers will independently screen studies, extract data, and assess the quality of included using the Newcastle-Ottawa Scale (NOS). We will harmonize data to calculate pooled relative risks (RRs) with 95% confidence intervals (CIs). Random-effects model is anticipated for high-heterogeneous results, while fixed-effects model will be adopted for low- heterogeneous results. Sub-group analyses (e.g., population type (general/occupational), geographic region, year of publication), sensitivity analyses, and assessments of publication bias (Egger's test, funnel plots) will be conducted to ensure the robustness and reliability of the findings. The analyses will be conducted via RevMan 5.3 and STATA 15 software.

**Data availability statement:** No datasets were generated or analysed during the current study. All relevant data from this study will be made available upon study completion.

**Funding:** The author(s) received no specific funding for this work.

**Competing interests:** The authors have declared that no competing interests exist.

## Discussion

This protocol will guide the standard process of the systematic review, providing synthesized evidence on the short-, and long-term effect of cadmium on lung risk in general and occupational populations.

## Conclusion

This systematic review and meta-analysis will provide synthesized evidence regarding cadmium exposure with lung cancer risk in both occupational and general populations. The evidence obtained in this study can inform the public and the policy-making bodies.

## Register on PROSPERO

CRD42024527248.

---

## Introduction

Lung cancer (LC) is the most common cancers and the leading cause of cancer-related death worldwide, causing a substantial economic burden at global, national, and individual levels [1–3]. The total death toll caused by lung cancer according to the Global cancer statistics 2022 was more than 18 million [3]. The lifetime risk of lung cancer was estimated at 6.7 (1/15) in male while 6.0 (1/17) in female [4]. While cigarette smoking remains the most important determinant of lung cancer, growing evidence suggests that environmental factors, e.g., heavy metals exposure could play an important role in the etiology and pathogenesis of lung cancer [5]. Heavy metals such as cadmium, can accumulate in lung tissue and exert carcinogenic effects [5].

Cadmium (Cd), a commonly found heavy metal in industrial society, is contained in soil, rocks, coal, and mineral fertilizers, etc. [6]. Its primary route of human exposure is through inhalation and ingestion, posing significant health risks to human owing to its long biological half-life [6]. Epidemiological studies have reported correlation between chronic cadmium exposure and an increased risk of various adverse health outcomes, including both cancer incidence and mortality [7–9]. Cadmium is classified as a carcinogen to human by the International Agency for Research on Cancer (IARC). It is primarily associated with lung cancer, as well as prostate, breast, and kidney cancers [10]. Despite being recognized as cancer carcinogen, and a few systematic review and meta-analyses have been conducted, whether mild exposure to cadmium will increase lung cancer risk and mortality is still uncertain [7,11-12–]. However, the evidence is still lacking regarding cadmium exposure in general population, who may be exposed to mild-to-moderate cadmium [7,11,12].

In this study, we will search all available studies to provide a comprehensive and up-to-date perspective on this topic. Our systematic review and meta-analysis is designed to fill the previous meta-analyses' gap by broader inclusion of cadmium measurements and specific sub-group designed for detailed assessment and analysis.

## Methods

### Study protocol

The study protocol was registered on the website of PROSPERO (No. CRD42024527248). This systematic review and meta-analysis will report following Preferred Reporting Items for Systematic Reviews and Meta-Ananlyses (PRISMA) guidelines [13]. (S1 File) https://www.crd.york.ac.uk/prospero/display_record.php?RecordID=527248.

### Search strategy

The researcher (LC) searched the five international electronic databases (PubMed, Embase, Web of Science, Medline and Cochrane library) from the inception of database to 31th December 2024. The search strategies for each database were originally designed by the author (LC), and were peer reviewed by another researcher (MZ). The search strategy applied on PubMed will be 'cadmium'[MeSH Terms] OR 'cadmium'[All Fields], and 'lung neoplasms'[MeSH Terms] OR ('lung'[All Fields] AND 'neoplasms'[All Fields]) OR 'lung neoplasms'[All Fields] OR ('lung'[All Fields] AND 'cancer'[All Fields]) OR ('lung cancer'[All Fields]). Two investigators will then screen all studies to assess their eligibility. Additionally, reference lists of all included studies will be searched to identify additional potentially eligible reports. Otherwise, there will be no additional information source for search. Filters were applied to limit the studies focusing on human beings and were published by 2024. Detailed information of search strategy can be accessible in the S2 File Search strategy. The checklist for reporting search strategy can be found in the S3 File PRISMA-S Checklist.

### Eligibility criteria and study selection

The inclusion criteria will be: a) observational human studies reporting cadmium exposure and lung cancer incidence and or mortality. b) The exposure of interest should include cadmium (can be blood, urinary, hair scalp, or other tissue samples), while the outcome of interest should include lung cancer. c) Studies reporting odds ratio (OR), relative risk (RR), hazard ratio (HR), or raw calculable data. d) Published studies without language restriction.

The exclusion criteria will be: a) reviews, systematic reviews, conference papers, letters, comments or case reports. b) Duplicates, or different studies that report almost the same cohort (the study that covered the largest population was chosen).

Two independent reviewers (LC, MZ) will screen and include the reports, while any conflict will be resolved by a third reviewer (SWQ). An inter-rater reliability is monitored to ensure the Cohen's $\kappa > 0.6$. The record screening is expected to be completed by the end of June. The flowchart of study selection process can be seen in Fig 1.

### Data extraction and quality assessment

The two authors will independently collect data on included studies. The information of first authors' last name, year of publication, region of the study, study design, the population of study (i.e., occupational, general, racial specific, or region-specific), the number of participants, the gender of participants if available, duration of follow-up (mean years, the year ends+, or person-years), levels of cadmium exposure, sample of cadmium exposure, measurements of the association, and adjusted covariates. The cadmium concentrations will be converted to μg/g creatinine for urine or μg/L for blood. For occupational population, the non-exposure or lowest quartile individuals will be regarded as reference group. Any discrepancy will be discussed and resolved after the third co-author, SWQ. Data extraction and subsequent results have not been conducted yet, but are expected to be completed by the end of June. The floor/ceiling effects will be addressed through $LOD/\sqrt{2}$ imputation for nondetectable cadmium levels and trim-and-fill analysis for extreme exposures. Quality assessment will apply the Newcastle-Ottawa Scale criteria. A score of 7–9 suggests high-quality, while the score of 4 or less indicates low quality. The moderate quality is in between. The summarized results will then be assessed following the evidence using the Grading of Recommendations, Assessment, Development, and Evaluations (GRADE) system [14].

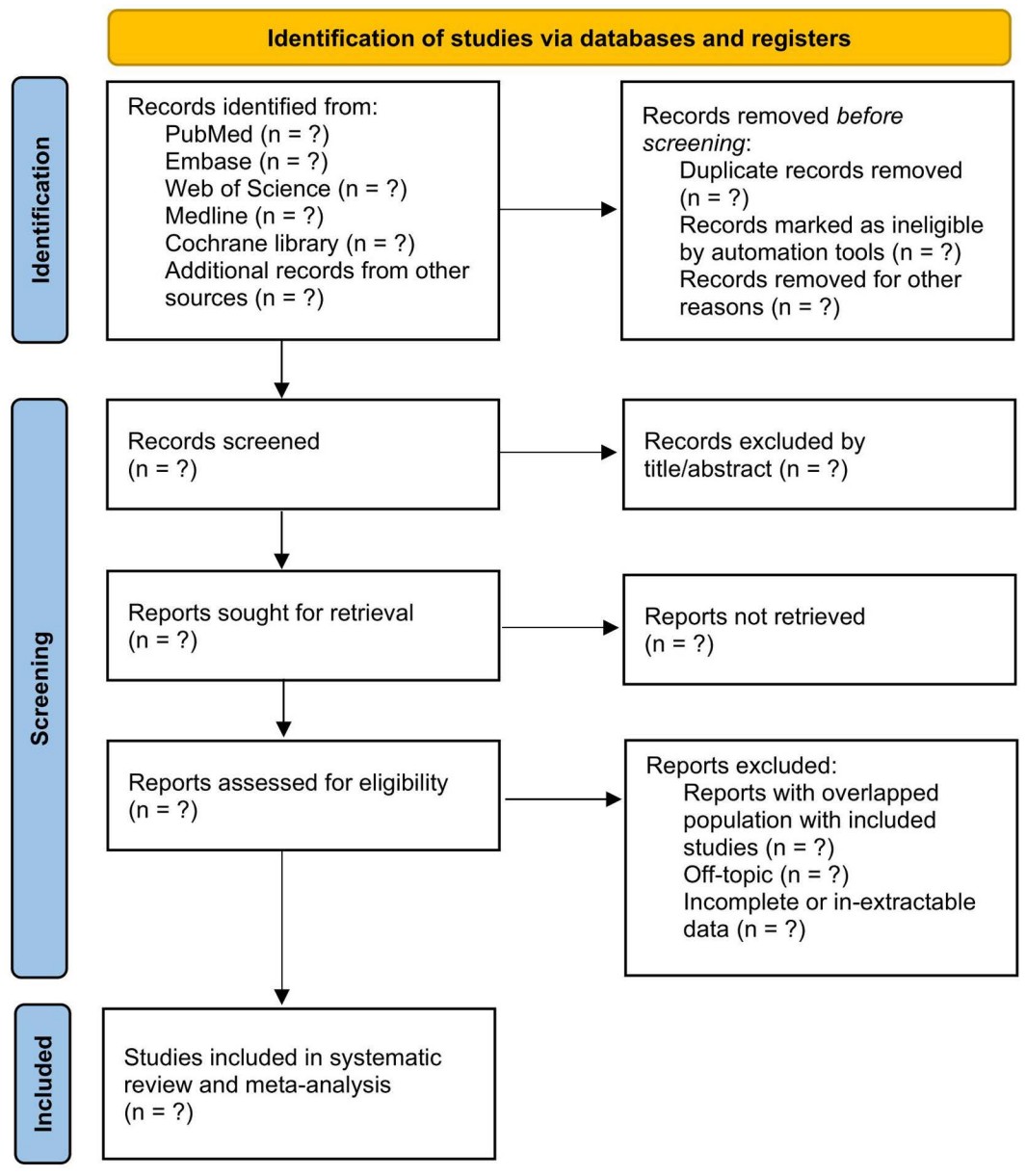

**Fig 1. Flow diagram of selection process of included studies.**

## Sub-group analyses

Sub-group analyses will be conducted to assess whether population difference, year of the research, smoking status, histological types of lung cancer, and gender could influence the results, or could be the source of heterogeneity. Specific population (racial specific, population from cadmium-polluted areas, or occupational population) will be extracted and compared with general population. Included studies will be further divided not according to the year of publication, but the time of the research as the lifestyle and working environment have changed greatly. As gender differences may also

influence lung cancer risk, we will extract data that separately reported lung cancer risks in men and women, and calculate their pooled relative risks respectively.

## Statistical analysis

The statistical analysis will be conducted using standardized instruments from the Cochrane Handbook or the JBI manual [15,16]. Weighted, adjusted RR/OR will be used to assess the correlation between cadmium exposure and lung cancer risk and/or mortality. In the overall results, OR will be converted to RR following the equation: $RR = OR/((1 - P_0) + (P_0 * OR))$ [17]. HR will be considered as RR in our analysis. We will adopt sensitivity analyses will evaluate these assumptions. Heterogeneity across studies will be assessed by $I^2$ (≥ 75% suggested high heterogeneity) [18]. Pre-specify controls will be applied for minimizing bias, such as smoking status, age, occupational co-exposures) as a key source of heterogeneity. Studies failing to adjust for ≥3 major confounders (smoking, age, sex) will be excluded in sensitivity analyses. The forest plot will be used to visualize the overall results. Either random-effects or fixed-effects models will be used depending on the heterogeneity between studies. Sensitivity validations will be employed to exclude studies exceeding 20% cumulative incidence for HR to RR conversion and Zhang and Yu adjustment for OR to RR where control-group prevalence is reported [19]. Additionally, the leave-one-out sensitivity analysis will be conducted to evaluate whether any single study could impact on the overall results. Publication bias will be conducted via funnel plot and further assessed by Egger's test (not applicable if included studies < 10) [20]. The analyses will be conducted via RevMan 5.3 and STATA 15 software.

## Discussion

This systematic review and meta-analysis will comprehensively assess and synthesize the most up-to-date evidence on the association between cadmium exposure and lung cancer risk. The results of the systematic review and meta-analysis will provide an integrated understanding of cadmium's potential carcinogenic effects across various exposure settings, particularly insights in both the general and occupational population. The findings of this review are expected to offer valuable insights for public health policy, risk assessment, and future research priorities.

If a positive association between cadmium exposure and general/occupational population risk of lung cancer is confirmed, several plausible pathways have been proposed. Firstly, cadmium-induced genotoxicity could lead to DNA damage, or epigenetic alterations through the generation of reactive oxygen species (ROS), interfering with DNA repair mechanisms, or DNA methylation, or histone acetylation [21]. Chronic exposure to cadmium can lead to the accumulation of genetic mutations in lung cells, potentially triggering oncogenic processes and promoting the development of lung cancer. Cadmium-induced epigenetic changes may dysregulate the expression of genes involved in oncogenesis and tumor suppression, potentially predisposing lung cells to malignancy [22]. If the association is not validated following strict conduction of synthesis of evidence, the cadmium will continue to be an essential contributing factor for developing lung cancer in occupational populations [12].

There were three previous meta-analysis have attempted to evaluate the correlation between cadmium exposure and lung cancer, the relationship, but each had notable limitations [7,11-12–]. Chen et al. calculated the risks of general and occupational population separately, marked different gender in the group analysis [12]. However, due to limited research at that time, they did not obtain definite conclusions either in general or occupational population. Nawrot et al. study evaluated cadmium exposure and lung cancer risks with only three prospective cohorts included for analysis [11]. Fanfani et al. conducted a comprehensive meta-analysis on the association between cadmium exposure and cancers' incidence and mortality of different sites, such as lung, prostate, thyroid, etc. [7]. Their reports were comprehensive, but they missed a few records when compared to Chen's in the same scope [12]. This may have influenced the validity and completeness of their pooled estimates. These methodological differences underscore the need for a more up-to-date and methodologically robust meta-analysis focused specifically on lung cancer.

The strengths of our study lie in its comprehensive approach to exposure assessment and rigorous sub-group analyses. Secondly, in contrast to earlier studies, our meta-analysis will include all relevant biological matrices for cadmium measurements, such as tissue, blood, urine, and hair scalp. This will enhance the comprehensiveness and precision of exposure classification, thereby improving the quality of the synthesized evidence. Thirdly, the sub-group analysis will we plan to conduct detailed sub-group analyses based on population type (general versus occupational), gender, ethnic, geographic region, study design, and histological types of lung cancers. These analyses will help to clarify sources of heterogeneity and identify the populations most vulnerable to cadmium exposure. This stratified approach is essential for accurately characterizing risk patterns and informing targeted interventions.

Nonetheless, this study may face several limitations. First, significant between-study heterogeneity is anticipated due to differences in study populations, exposure assessment methods, confounding control, and outcome definitions. Such heterogeneity could influence the reliability and interpretability of the synthesized evidence. To address the pivotal challenges, authors will conduct sensitivity analysis include sub-group analysis for possible heterogeneous causes, leave-one-out analysis, funnel plot, and Egger's test if more than ten studies will be included. Second, variability in follow-up duration across studies may impact the observed associations. Given the long latency period of lung cancer, studies with short-term follow-up may underestimate the true risk associated with chronic cadmium exposure. Third, residual confounding, especially from smoking and occupational co-exposures, may persist in some included studies, despite adjustments. We will address these challenges through sensitivity analyses and rigorous quality assessment.

Future research should prioritize prospective cohort studies with long-term follow-up to better capture the latency of cadmium-induced carcinogenesis. Moreover, establishing a dose-response relationship between cadmium levels and lung cancer incidence and mortality will be crucial for refining exposure thresholds and public health guidelines.

In summary, this meta-analysis will contribute to filling existing knowledge gaps by providing a systematic and comprehensive evaluation of the current evidence on cadmium exposure and lung cancer risk. The results will have implications for environmental health regulation, occupational safety standards, and preventive strategies against cadmium-related lung carcinogenesis.

## Supporting information

**S1 File. The PRISMA-P checklist for this systematic review protocol.**
(DOCX)

**S2 File. Details of the search strategy for each database.**
(DOCX)

**S3 File. The PRISMA-S checklist for the reporting of the search strategy for this systematic review protocol.**
(DOCX)

## Author contributions

**Conceptualization:** Lin Chen, Shuiwei Qiu.

**Data curation:** Lin Chen, Min Zhou, Shuiwei Qiu.

**Formal analysis:** Lin Chen, Dingliang Lv.

**Investigation:** Lin Chen, Min Zhou, Shuiwei Qiu.

**Methodology:** Min Zhou, Dingliang Lv.

**Resources:** Min Zhou, Dingliang Lv.

**Software:** Lin Chen, Min Zhou.

**Supervision:** Dingliang Lv, Shuiwei Qiu.

**Validation:** Lin Chen, Min Zhou, Shuiwei Qiu.

**Visualization:** Lin Chen, Dingliang Lv.

**Writing – original draft:** Lin Chen, Min Zhou, Shuiwei Qiu.

**Writing – review & editing:** Lin Chen, Shuiwei Qiu.

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
