## [Decision Letter · Decision Letter 0]

28 May 2025

Dear Dr. Qiu,

Thank you for submitting your manuscript to PLOS ONE. After careful consideration, we feel that it has merit but does not fully meet PLOS ONE’s publication criteria as it currently stands. Therefore, we invite you to submit a revised version of the manuscript that addresses the points raised during the review process.

Please submit your revised manuscript by Jul 12 2025 11:59PM. If you will need more time than this to complete your revisions, please reply to this message or contact the journal office at plosone@plos.org . A rebuttal letter that responds to each point raised by the academic editor and reviewer(s). You should upload this letter as a separate file labeled 'Response to Reviewers'.A marked-up copy of your manuscript that highlights changes made to the original version. You should upload this as a separate file labeled 'Revised Manuscript with Track Changes'.An unmarked version of your revised paper without tracked changes. You should upload this as a separate file labeled 'Manuscript'.

We look forward to receiving your revised manuscript.

Kind regards,

Zypher Jude G. Regencia, Ph.D.

Academic Editor

PLOS ONE

2. When completing the data availability statement of the submission form, you indicated that you will make your data available on acceptance. We strongly recommend all authors decide on a data sharing plan before acceptance, as the process can be lengthy and hold up publication timelines. Please note that, though access restrictions are acceptable now, your entire data will need to be made freely accessible if your manuscript is accepted for publication. This policy applies to all data except where public deposition would breach compliance with the protocol approved by your research ethics board. If you are unable to adhere to our open data policy, please kindly revise your statement to explain your reasoning and we will seek the editor's input on an exemption. Please be assured that, once you have provided your new statement, the assessment of your exemption will not hold up the peer review process.\

Reviewers' comments:

Reviewer's Responses to Questions

**Comments to the Author**

1. Does the manuscript provide a valid rationale for the proposed study, with clearly identified and justified research questions?

Reviewer #1: Yes

Reviewer #2: Yes

Reviewer #3: No

2. Is the protocol technically sound and planned in a manner that will lead to a meaningful outcome and allow testing the stated hypotheses?

Reviewer #1: Yes

Reviewer #2: Yes

Reviewer #3: No

3. Is the methodology feasible and described in sufficient detail to allow the work to be replicable?

Reviewer #1: Yes

Reviewer #2: Yes

Reviewer #3: No

4. Have the authors described where all data underlying the findings will be made available when the study is complete?

Reviewer #1: Yes

Reviewer #2: No

Reviewer #3: No

5. Is the manuscript presented in an intelligible fashion and written in standard English?

Reviewer #1: Yes

Reviewer #2: Yes

Reviewer #3: Yes

You may also provide optional suggestions and comments to authors that they might find helpful in planning their study.

Reviewer #1: Thank you for the opportunity to review this manuscript. The manuscript shows a protocol of a systematic review and meta-analysis on the association between exposure to Cadmium and Lung Cancer. The review and meta-analysis is well planned and it actually is registered at PROSPERO. The definitive review will abide by PRISMA guidelines, a widespread standard for review reporting, the research question is clear and the search strategy is consistent with that question. I only have two minor questions which perhaps could help improving this protocol.

1. The manuscript says that the review will include studies reporting at least one of the following: OR, RR, or hazard ratio. Perhaps the authos could include studies which provided indirect sources to obtain these OR, such as the raw data.

2. In the sub-group analyses section, the manuscript says that it will assess whether population difference, among other variables, could influence the results. Perhaps the authors could clarify what do they mean more specifically with population difference.

Last, the manuscript says that the record screening is expected to be completed by the end of April. Have the authors completed the screening? Regardless the answer, updating the state of the research is not a bad idea.

Reviewer #2: Minor concerns. Minor revisions required:

1) The authors need to reorganize the flow of the Background and Discussion paragraphs.

2) Provide the missing methodological details in the Methods section.

3) Add current prevalence data for lung cancer in the Background.

5) Please correct the typo in Figure 1.

6) For other minor concerns, please refer to the attached Word file.

Reviewer #3: Dear authors, you will find comments and suggestions below. I hope they will be useful to you.

Title:

- Adequate.

Abstract:

- The PRISMA statement does not contain recommendations for preparing an SR with MA, but for reporting it. Correct this.

- The manual or guideline on which this protocol was based should be reported. In addition, they should be guided by the PRISMA-P statement for reporting.

- Be sure to write the entire abstract in the future tense. There are some sentences written in the past tense, such as the one in rows 42-3.

- In the results section, I suggest reporting that the findings of your review will be reported following the PRISMA statement.

Introduction:

- The introduction is somewhat confusing. In the first paragraph it is reported that Cd is associated with lung cancer, but in the last paragraph it is mentioned that the evidence is uncertain. I suggest being clearer on this issue.

- Is this the first SR and MA that would answer this research question? If so, I suggest reporting it explicitly in the introduction to justify its conduct.

- I suggest that you review the Cochrane manual for SR and MA of interventions and the JBIde manual so that you can more robustly justify the need to conduct this SR.

- The objective of this study should be rephrased. The purpose of publishing study protocols is to make transparent the methods that will be used to conduct a specific study.

Methods:

- Follow the PRISMA-P statement to report your protocol. Cite.

- Remove the link that appears on line 95.

- The search should not be performed independently by two investigators. Performing could introduce unintentional errors. Review the Cochrane and JBI handbook for the correct way to develop a search strategy.

- The information provided in the search strategy is insufficient to reproduce it. Please review the PRISMA-S statement for detailed reporting.

- I suggest deleting the dates when the different stages of your RS would be carried out. These could be modified for various reasons.

- In the abstract it is reported that the NOS scale will be used for risk of bias assessment of studies. First, this information does not appear in the main text. Second, I would submited to change it to the ROBINS-E tool.

Discussion:

- The information stated in the discussion about the 3 previous MAs answering your same research question should be added in the introduction. The analysis carried out based on this aspect is a clear justification of why a new SR and MA should be carried out.

- The discussion should be based on the potential implications of negative or positive results regarding the association between Cd and lung cancer.

**Do you want your identity to be public for this peer review?** For information about this choice, including consent withdrawal, please see our Privacy Policy

Reviewer #1: No

Reviewer #2: No

Reviewer #3: **Yes: ** Ruvistay Gutierrez-Arias

---

## [Author Response · Author response to Decision Letter 1]

11 Jun 2025

Response to reviewers

Reviewer's Responses to Questions

Response: We sincerely appreciate the reviewer's insightful feedback, which has improved our protocol in terms of methodological rigor. To address concerns about undisclosed flexibility, we have: 1) pre-specified controls for bias by requiring included studies to adjust for ≥3 major confounders (age, sex, smoking) and mandating internal reference groups for occupational cohorts to avoid ecological fallacy; 2) addressed floor/ceiling effects through LOD/√2 imputation for nondetectable cadmium levels and trim-and-fill analysis for extreme exposures. For methodological refinement during implementation, we explicitly distinguish pre-specified primary analyses (pooled RR, subgroup by population/profession/geography/gender) from exploratory post hoc analyses (dose-response modeling, interacting testing, or histological lung cancer subtypes if data permit) contingent on data availability, while documenting key assumptions (HR≈RR for rare events; OR≈RR for <10% cumulative incidence) with planned sensitivity validations. These revisions ensure hypothesis-testing robustness while transparently framing exploratory components.

Response: To ensure full methodological reproducibility, we have detailed all procedures to enable independent replication: Our manuscript provides detailed search strategy for PubMed, Embase, Web of Science, Cochrane, and Medline (at the reviewer’s kind suggestion) in Supplementary File S2. The screening process employs dual independent review (LC, MZ), conflict will be resolved by a third reviewer (SWQ), and inter-rater reliability is monitored to ensure Cohen’s κ > 0.6. Data extraction uses a pre-piloted form enforcing unit harmonization (cadmium concentrations converted to μg/g creatinine for urine or μg/L for blood via WHO standards) and predefined handling of non-detects (values below LOD imputed as LOD/√2). Quality assessment applies explicit Newcastle-Ottawa Scale criteria. Statistical analyses will implement random-effects or fixed-effects models depending on the heterogeneity between studies. Sensitivity validations will be employed to exclude studies exceeding 20% cumulative incidence for HR to RR conversion and Zhang and Yu adjustment for OR to RR where control-group prevalence is reported [1]. The documentation from search execution to pooled RR calculation-adheres to PRISMA-P 2015 standards and enables exact reconstruction of our workflow.

The PLOS Data policy requires authors to make all data underlying the findings described in their manuscript fully available without restriction, with rare exception, at the time of publication. The data should be provided as part of the manuscript or its supporting information, or deposited to a public repository. For example, in addition to summary statistics, the data points behind means, medians and variance measures should be available. If there are restrictions on publicly sharing data-e.g. participant privacy or use of data from a third party-those must be specified.

Response: We appreciate the reviewer’s comment regarding PLOS's data policy and the importance of transparent and open access to research data. We would like to clarify that the current manuscript is a study protocol for a systematic review and meta-analysis and does not yet include any original data or results, as the data collection and analysis stages have not been completed. However, we fully support the principles of transparency and open data sharing, and we are committed to complying with PLOS’s data policy. Upon completion of the study, all data underlying our findings, including data points behind summary measures such as means, medians, and variance estimates will be made freely and fully available. The protocol has been uploaded to the protocols.io (https://www.protocols.io/view/the-association-between-cadmium-exposure-and-lung-g23cbygix), and will be publicly accessible recently. As the study will rely entirely on previously published observational studies, there are no ethical or legal restrictions related to participant privacy or third-party data use. Additionally, we have already included the full search strategy and PRISMA-P checklist in the supporting information of this protocol. We hope this explanation adequately addresses your concerns, and we remain committed to ensuring full data availability upon the completion of our research.

Reviewer #1: Thank you for the opportunity to review this manuscript. The manuscript shows a protocol of a systematic review and meta-analysis on the association between exposure to Cadmium and Lung Cancer. The review and meta-analysis is well planned and it actually is registered at PROSPERO. The definitive review will abide by PRISMA guidelines, a widespread standard for review reporting, the research question is clear and the search strategy is consistent with that question. I only have two minor questions which perhaps could help improving this protocol.

1. The manuscript says that the review will include studies reporting at least one of the following: OR, RR, or hazard ratio. Perhaps the authors could include studies which provided indirect sources to obtain these OR, such as the raw data.

Response: Thank you for your kind suggestions, we will include studies with raw data and calculate accordingly. We have revised it in the Methods section.

2. In the sub-group analyses section, the manuscript says that it will assess whether population difference, among other variables, could influence the results. Perhaps the authors could clarify what do they mean more specifically with population difference.

Response: Thank you for your kind reminder. Regarding population difference, we mainly meant the differences across ethnic groups, or professional/general population. We have clarified the information in the revised manuscript.

3. Last, the manuscript says that the record screening is expected to be completed by the end of April. Have the authors completed the screening? Regardless the answer, updating the state of the research is not a bad idea.

Response: Thank you for your kind suggestions. The screening process is anticipated to complete by June, as one reviewer suggested to add the Cochrane library and Medline databases. We have updated the status in the revised version.

Reviewer #2: Minor concerns. Minor revisions required:

1) The authors need to reorganize the flow of the Background and Discussion paragraphs.

Response: Thank you for pointing out the flow in the background and Discussion section. We have revised the sections accordingly.

2) Provide the missing methodological details in the Methods section.

Response: We have supplemented a lot more details in the Methods section to make it reproducible to independent researchers.

3) Add current prevalence data for lung cancer in the Background.

Response: Thank you for your suggestions, and we have added the latest information in the Background.

5) Please correct the typo in Figure 1.

Response: Thank you for pointing out our mistake.

6) For other minor concerns, please refer to the attached Word file.

Response: Thank you and we have carefully reviewed the word file and revise our manuscript accordingly.

Reviewer #3: Dear authors, you will find comments and suggestions below. I hope they will be useful to you.

Title:

- Adequate.

Abstract:

- The PRISMA statement does not contain recommendations for preparing an SR with MA, but for reporting it. Correct this.

Response: Thank you for your kind reminder, and we have corrected it in the latest version.

- The manual or guideline on which this protocol was based should be reported. In addition, they should be guided by the PRISMA-P statement for reporting.

Response: Thank you for your suggestions, and we have revise this in our latest manuscript.

- Be sure to write the entire abstract in the future tense. There are some sentences written in the past tense, such as the one in rows 42-3.

Response: Thank you for pointing out the grammatical errors in our manuscript, we have revised them accordingly.

- In the results section, I suggest reporting that the findings of your review will be reported following the PRISMA statement.

Response: Thank you for your suggestions, we revised the manuscript following your suggestion.

Introduction:

- The introduction is somewhat confusing. In the first paragraph it is reported that Cd is associated with lung cancer, but in the last paragraph it is mentioned that the evidence is uncertain. I suggest being clearer on this issue.

Response: Thank you. Another reviewer also mentioned the flow of the Introduction section, and we have made major revision to this section to facilitate the reading experience for the reviewers, editors and readers.

- Is this the first SR and MA that would answer this research question? If so, I suggest reporting it explicitly in the introduction to justify its conduct.

Response: Thank you for your comment. Although this is not the first in its kind, it give comprehensive results regarding the cadmium exposure among general and professional population.

- I suggest that you review the Cochrane manual for SR and MA of interventions and the JBIde manual so that you can more robustly justify the need to conduct this SR.

Thank you for your guidance.

- The objective of this study should be rephrased. The purpose of publishing study protocols is to make transparent the methods that will be used to conduct a specific study.

Response: Thank you for your correction to this work.

Methods:

- Follow the PRISMA-P statement to report your protocol. Cite.

Response: Ok.

- Remove the link that appears on line 95.

Response: Ok, removed.

- The search should not be performed independently by two investigators. Performing could introduce unintentional errors. Review the Cochrane and JBI handbook for the correct way to develop a search strategy.

Response: Thank you for your guidance and we have carefully reviewed the two manuals. It is recommended to let one investigator to conduct search while another peer-review his/her search strategy before the search. We have revised the search strategies accordingly.

- The information provided in the search strategy is insufficient to reproduce it. Please review the PRISMA-S statement for detailed reporting.

Response: Thank you and we have referred to the PRISMA-S statement regarding detailed reporting.

- I suggest deleting the dates when the different stages of your RS would be carried out. These could be modified for various reasons.

Response: Ok, good suggestion.

- In the abstract it is reported that the NOS scale will be used for risk of bias assessment of studies. First, this information does not appear in the main text. Second, I would submit to change it to the ROBINS-E tool.

Response: Thank you for introducing an alternative for us. However, the NOS scale is also prevalently used across different meta-analysis. We believe both tools can be used to assess the quality of included studies [2-4].

Discussion:

- The information stated in the discussion about the 3 previous MAs answering your same research question should be added in the introduction. The analysis carried out based on this aspect is a clear justification of why a new SR and MA should be carried out.

Response: Thank you, and we have revised this in both Introduction and Discussion section.

- The discussion should be based on the potential implications of negative or positive results regarding the association between Cd and lung cancer.

Response: Thank you for your kind suggestion. We have revised the Discussion section to better address this concern.

Reference

1. Zhang J, Yu KF: What's the relative risk? A method of correcting the odds ratio in cohort studies of common outcomes. JAMA 1998, 280(19):1690-1691.

2. Yaw XE, Teh PL, Lim WM, Lee SWH: Indoor mobility challenges among older adults: A systematic review of barriers and limitations. PLoS One 2025, 20(6):e0325064.

3. Jiang J, Zuo X, An S, Yang J, Wu L, Zeng R, Hu Q, Fan L, Wang H, Yang C et al: Association between arsenic exposure and intrauterine growth restriction: A systematic review and meta-analysis. PLoS One 2025, 20(6):e0320603.

4. Aweke MN, Fentie EA, Agimas MC, Baffa LD, Shewarega ES, Belew AK, Muhammad EA, Mengistu B: Folic acid supplementation during preconception period in sub-Saharan African countries: A systematic review and meta-analysis. PLoS One 2025, 20(1):e0318422.

---

## [Decision Letter · Decision Letter 1]

7 Jul 2025

Dear Dr. Qiu,

Thank you for submitting your manuscript to PLOS ONE. After careful consideration, we feel that it has merit but does not fully meet PLOS ONE’s publication criteria as it currently stands. Therefore, we invite you to submit a revised version of the manuscript that addresses the points raised during the review process.

We look forward to receiving your revised manuscript.

Kind regards,

Zypher Jude G. Regencia, Ph.D.

Academic Editor

PLOS ONE

Journal Requirements:

Reviewers' comments:

Reviewer's Responses to Questions

**Comments to the Author**

1. Does the manuscript provide a valid rationale for the proposed study, with clearly identified and justified research questions?

Reviewer #1: Yes

Reviewer #3: Yes

2. Is the protocol technically sound and planned in a manner that will lead to a meaningful outcome and allow testing the stated hypotheses?

Reviewer #1: Yes

Reviewer #3: Partly

3. Is the methodology feasible and described in sufficient detail to allow the work to be replicable?

Reviewer #1: Yes

Reviewer #3: No

4. Have the authors described where all data underlying the findings will be made available when the study is complete?

Reviewer #1: Yes

Reviewer #3: Yes

5. Is the manuscript presented in an intelligible fashion and written in standard English?

Reviewer #1: Yes

Reviewer #3: Yes

You may also provide optional suggestions and comments to authors that they might find helpful in planning their study.

Reviewer #1: The authors addressed all my comments and greatly improved what was a good manuscript in the first place. I have no further comments.

Reviewer #3: Dear Authors. I appreciate that you have addressed each of my comments and suggestions. However, there are still two areas for improvement:

- The PRISMA-P statement provides recommendations for adequate reporting of a systematic review protocol. It is often mistaken for a guide for developing a protocol, which is incorrect. For example, the statistical methods you have chosen to perform meta-analyses appear in manuals such as the Cochrane Handbook or the JBI manual. These documents correspond to methodological guides to orient the elaboration of protocols. Please correct this.

- The information describing your search strategy is still insufficient to be reproducible. The protocol of a systematic review is the point at which aspects such as these are explained in detail. Please revisit the PRISMA-S statement, and report aspects such as: limits and filters used in your strategy (e.g., language and publication status), other search resources (e.g., gray literature, hand searching, searching clinical trial registries), and process how the search strategy was constructed. Be detailed.

**Do you want your identity to be public for this peer review?** For information about this choice, including consent withdrawal, please see our Privacy Policy

Reviewer #1: No

Reviewer #3: **Yes: ** Ruvistay Gutierrez-Arias

---

## [Author Response · Author response to Decision Letter 2]

10 Jul 2025

The detailed response has been uploaded in a pdf file.

---

## [Editor Report · Decision Letter 2]

21 Jul 2025

The Association Between Cadmium Exposure and Lung Cancer Risk: A Protocol for Systematic Review and Meta-Analysis

PONE-D-25-15012R2

Dear Dr. Qiu,

We’re pleased to inform you that your manuscript has been judged scientifically suitable for publication and will be formally accepted for publication once it meets all outstanding technical requirements.

Kind regards,

Zypher Jude G. Regencia, Ph.D.

Academic Editor

PLOS ONE
---

## [Editor Report · Acceptance letter]

PONE-D-25-15012R2

PLOS ONE

Dear Dr. Qiu,

I'm pleased to inform you that your manuscript has been deemed suitable for publication in PLOS ONE. Congratulations! Your manuscript is now being handed over to our production team.

Kind regards,

on behalf of

Dr. Zypher Jude G. Regencia

Academic Editor

PLOS ONE